# Shake It Off: The Elimination of Erroneous Kinetochore-Microtubule Attachments and Chromosome Oscillation

**DOI:** 10.3390/ijms22063174

**Published:** 2021-03-20

**Authors:** Ayumu Yamamoto

**Affiliations:** Graduate School of Integrated Science and Technology, Shizuoka University, 836 Ohya, Suruga-ku, Shizuoka 422-8529, Japan; yamamoto.ayumu@shizuoka.ac.jp; Tel.: +81-54-238-4762

**Keywords:** spindle, kinetochore, microtubule, chromosome oscillation, Aurora B kinase, tension, centromere

## Abstract

Cell proliferation and sexual reproduction require the faithful segregation of chromosomes. Chromosome segregation is driven by the interaction of chromosomes with the spindle, and the attachment of chromosomes to the proper spindle poles is essential. Initial attachments are frequently erroneous due to the random nature of the attachment process; however, erroneous attachments are selectively eliminated. Proper attachment generates greater tension at the kinetochore than erroneous attachments, and it is thought that attachment selection is dependent on this tension. However, studies of meiotic chromosome segregation suggest that attachment elimination cannot be solely attributed to tension, and the precise mechanism of selective elimination of erroneous attachments remains unclear. During attachment elimination, chromosomes oscillate between the spindle poles. A recent study on meiotic chromosome segregation in fission yeast has suggested that attachment elimination is coupled to chromosome oscillation. In this review, the possible contribution of chromosome oscillation in the elimination of erroneous attachment is discussed in light of the recent finding.

## 1. Introduction

During mitotic cell division, replicated chromosomes termed sister chromatids become attached to opposite spindle poles (referred to as bi-oriented attachment) and separate from each other (equational segregation), leading to the formation of two genetically identical daughter cells (Figure 1a, mitosis) [1]. During the first division of germ cells (meiosis I), homologous chromosomes attach to opposite spindle poles and separate from each other (reductional segregation) (Figure 1a, meiosis I). Subsequently, sister chromatids undergo equational segregation (meiosis II), resulting in the formation of haploid gametes. In both equational and reductional segregations, chromosomes interact with the spindle in a random manner initially, resulting in the frequent attachment of chromosomes to improper poles (Figure 1b) (reviewed in References [2,3]). These erroneous attachments are eliminated, and the chromosomes reattach to the spindle. Each chromosome repeats the attachment and detachment processes, until it becomes properly attached to the spindle; however, the mechanism by which improper attachments are eliminated is not fully understood.

Chromosomes interact with the spindle at the kinetochore, a protein complex assembled on the centromere (Figure 1a). It is widely believed that the selection of proper attachments is dependent on tension generated at the kinetochore [2,3,4,5,6]. During mitosis, when sister chromatids are correctly bi-oriented and pulled toward opposite poles, tension is generated at the kinetochores as sister chromatids are held together by a protein complex called cohesin [7]. However, when the sister chromatids become improperly attached to the same pole (referred to as mono-oriented attachment), sufficient tension is not generated. Likewise, during meiosis I, the bi-oriented attachment of homologous chromosomes generates tension, as homologous chromosomes are linked by a recombination product called chiasma, while erroneous mono-oriented attachments do not. It has been proposed that the tension causes the stabilization of otherwise unstable, and readily eliminated, attachments [2,4]. This model is supported by the observation that mono-oriented homologous chromosomes remain attached to the spindle at meiosis I when anti-poleward forces are exerted on the chromosomes with an ultra-fine needle (Figure 1b, meiosis I) [4]. This evidence is compelling, and tension-dependent attachment stabilization has been the underlying principle guiding current research into chromosome segregation.

However, the elimination of erroneous attachment may be more complicated. In meiosis I, the establishment of tension-dependent attachments does not fully account for attachment selection. In addition, live-cell imaging has revealed that, following the interaction of the kinetochores with spindle microtubules (MTs), chromosomes repeatedly oscillate between the spindle poles until chromosomes undergo segregation [8,9,10,11]. Despite the concomitant occurrence, attachment elimination has not been considered in the context of chromosome oscillation in many published studies, and the relationship between elimination of attachments and chromosome oscillation remains unclear. In this review, I introduce a recent study on attachment establishment during meiosis I in fission yeast, which suggests a link between chromosome oscillation and the elimination of erroneous attachments [12], and I discuss the possible contribution of chromosome oscillation in attachment elimination.

## 2. Tension-Dependent Attachment Establishment and Chromosome Dynamics during the Establishment

First, a brief overview of the mechanisms of tension-dependent attachment establishment and chromosome oscillation is described. Their mechanisms have been reviewed in greater detail elsewhere [3,6,10,11,13,14,15,16,17,18,19].

### 2.1. The Current Model for the Tension-Dependent Attachment Establishment

In the current model, Aurora B kinase, which is enriched at the inner kinetochore/centromere region underneath the outer kinetochore, is thought to play a central role in the tension-dependent establishment of attachments. Aurora B weakens kinetochore–MT interaction by phosphorylating kinetochore components, and it has been proposed that attachment establishment is dependent on tension-dependent spatial separation of attachment sites from the Aurora B–enriched region [15,20,21,22,23,24,25] (Figure 2). Under tension, kinetochore–MT attachment sites are spatially separated from the Aurora B–enriched region, and this tension-dependent separation, together with the action of PP1 phosphatase [26,27,28,29], impedes Aurora B–dependent phosphorylation of kinetochore components, stabilizing the attachment (Figure 2, bi-oriented). By contrast, the proximity of tensionless attachment sites to the Aurora B–enriched region leads to phosphorylation of kinetochore components, releasing kinetochore-interacting MTs (kMTs) from the kinetochore (Figure 2, mono-oriented). In vertebrates, Aurora A kinase additionally contributes to kinetochore phosphorylation-dependent elimination of erroneous attachments [30,31].

In addition to erroneous mono-oriented attachment (all kinetochores attached at a single spindle pole), a single sister chromatid or sister chromatids become erroneously bi-oriented during mitosis or meiosis I, respectively (referred to as merotelic attachment) (Figure 2, merotelic) [32,33]. Elimination of these merotelic attachments can also be explained by the tension-dependent spatial separation model; merotelic attachments cause distortion and stretching of the kinetochore, resulting in the proximity of attachment sites to the Aurora B–enriched region and the subsequent elimination of attachments (Figure 2, merotelic) [20,23,34].

### 2.2. Chromosome Oscillation during Attachment Establishment

Kinetochores initially interact with the lateral surface of spindle MTs and then at the MT ends, resulting in bi-oriented attachment of chromosomes (Figure 3a) (e.g., see References [8,35,36]). Upon spindle interaction, chromosomes instantly start back-and-forth movements between the spindle poles, and these oscillatory movements continue until the chromosomes undergo segregation. Oscillatory movements of the chromosomes that are laterally interacting with the spindle MTs depend on the poleward and anti-poleward sliding of kinetochore-localized MT motors, such as kinesin and cytoplasmic dynein [36,37,38,39,40,41,42,43,44]. On the other hand, oscillatory movements of MT end-interacting chromosomes are driven by disassembly of kMTs, and the Ndc80 and Dam1/DASH kinetochore complexes (Ska complex in vertebrates) couple kMT disassembly with kinetochore movements [45,46,47,48,49,50,51,52,53,54,55,56,57,58,59,60,61,62,63,64,65,66]. In *S. pombe*, kinetochore-localized kinesin-8 motors also contribute to the kMT disassembly dependent kinetochore movements [64,67,68,69,70,71,72]. In metazoans, anti-poleward forces (polar ejection forces) generated by kinesin motor-dependent MT interactions with chromosome arms additionally aid chromosome oscillation [73,74,75,76,77,78].

The oscillation of MT end-interacting bi-oriented chromosomes requires the coordinated switching of assembly/disassembly dynamics of kMTs (reviewed in References [10,11]) (Figure 3b). During chromosome movement, leading kMTs shorten and move centromeres forward, while trailing kMTs elongate. Switching of assembly/disassembly dynamics of either the leading or trailing kMTs induces the coordinated switching of other kMTs and the reversal of centromere movements [79]. Various in vitro, in vivo, and in silico studies have demonstrated that initial switching is induced either stochastically or by MT length-dependent disassembly factors, and that changes in tension across the bi-oriented centromeres causes subsequent switching of other kMTs [8,70,71,72,80,81,82,83,84,85,86,87,88,89,90] (Figure 3b, transition state).

Like the establishment of chromosome attachment, chromosome oscillation is regulated by Aurora B kinase. It was shown that the introduction of an Ndc80 variant whose Aurora B phosphorylation sites were replaced with alanine inhibited chromosome oscillation but allowed poleward movement of centromeres [91,92]. This demonstrates that Aurora B–dependent kinetochore phosphorylation is required for chromosome oscillation but not for poleward chromosome movements. Consistently, Aurora B–dependent kinetochore phosphorylation is crucial during the chromosome oscillation stage but not during anaphase, where chromosomes move solely poleward (reviewed in Reference [13]). In support of the dispensability of Aurora B–dependent phosphorylation for poleward chromosome movements, in vitro studies have demonstrated that non-phosphorylated kinetochore components can follow disassembling MT ends and couple MT disassembly with cargo movements [49,61,93,94,95]. Aurora B–dependent phosphorylation weakens MT affinity of the kinetochore components and allows their bi-directional, diffusion-like sliding on MTs [61,93,94,96,97,98]. Perhaps, bi-directional sliding of the phosphorylated kinetochores allows chromosome oscillation by enabling kinetochores to follow both assembling and disassembling MT ends, but strong non-mobile interaction of the non-phosphorylated kinetochores would allow only MT disassembly–dependent poleward centromere movements (Figure 3c) [50].

## 3. Problems with the Tension-Dependent Spatial Separation Model during Meiosis

Although numerous studies have provided evidence that supports a tension-dependent spatial separation model, the mechanism by which attachment is established in meiosis remains unclear. In the fission yeast *Schizosaccharomyces pombe*, sister centromeres split upon bi-oriented attachment [99,100], enabling the sister kinetochores to readily face in opposite directions during meiosis I (Figure 4a). In this situation, the Aurora-enriched region would remain away from the attachment sites. Moreover, in mouse oocytes, tension generated at the kinetochores does not correlate with proper attachment, and attachment sites under tension are not spatially separated from the Aurora-enriched region [101]. Therefore, the tension-dependent spatial separation model does not fully account for the elimination of erroneous attachments during meiosis I.

## 4. The Intimate Relationship between Attachment Elimination and Centromere Oscillation

The very recent study by Wakiya et al. demonstrated the intimate relationship between elimination of erroneous attachment and centromere oscillation during meiosis I chromosome segregation in *S. pombe* [12]. It is well-known that the connection of homologous chromosomes by the chiasma is essential for the bi-oriented attachment of homologous chromosomes during meiosis I. However, it has also long been known that the chiasma is essential for mono-oriented attachment of sister chromatids [34,100,102,103,104,105]. It was found that the spindle assembly checkpoint (SAC) factor Mad2, which delays anaphase onset to correct erroneous attachment, and the *S. pombe* Aurora B kinase Ark1 are crucial for chiasma-dependent mono-oriented attachment of sister chromatids. In the presence of chiasmata, Mad2 and Ark1 promoted elimination of bi-oriented attachments of sister chromatids, whereas, in the absence of chiasmata, they promoted elimination of mono-oriented attachments. These results indicated that the chiasma changed attachment-correction patterns by enabling error-correction factors to eliminate the bi-oriented attachment of sister chromatids, which is otherwise not subject to elimination.

Although a decrease in tension is thought to cause attachment elimination, an examination of inter sister centromere distances failed to demonstrate a significant reduction in tension across sister centromeres in chiasma-forming cells and rather suggested that a subset of bi-oriented sister centromeres was under greater tension in the presence of chiasmata. A further investigation of centromere dynamics demonstrated that the chiasma coordinates homologous centromere oscillation. These observations raised the possibility that coordinated chiasma-dependent homologous centromere oscillation contributes to the elimination of bi-oriented attachments of sister chromatids. Analyses of cells lacking the kinetochore component Dam1 has provided evidence supporting this possibility. In cells containing a deletion of the *dam1* gene (*dam1∆*), correction of erroneous attachments was compromised, as observed in *mad2*- or *ark1*-mutant cells; bi-oriented attachment of sister chromatids increased in the presence of chiasmata, but it conversely decreased in the absence of chiasmata. Detailed analysis of centromere dynamics demonstrated that the *dam1∆* mutation did not compromise spindle elongation-dependent chromosome segregation (anaphase B) but completely abolished homologous centromere oscillation together with anaphase poleward centromere movements (anaphase A), both of which depend on kMT disassembly. The coincidental impairment of attachment correction and homologous centromere oscillation observed in *dam1∆* cells supported the possibility that chromosome oscillation contributes to attachment elimination.

## 5. The Possible Contribution of Centromere Oscillation to Attachment Elimination

By what mechanism could coordinated centromere oscillation contribute to attachment elimination? In chiasma-forming cells, when homologous chromosomes oscillate with one of sister chromatid pairs improperly bi-oriented, stochastic, and/or length-dependent switching of assembly/disassembly dynamics of kMTs perhaps gives assembling kMT ends a chance to experience a chiasma-dependent minus end-directed load specifically at improper attachment sites (Figure 4b, meiosis I, merotelic). Wakiya et al. proposed that the application of the minus end-directed load to assembling kMTs results in their detachment [12]. Detachment of kMTs may result from bringing of the kMT ends to the Aurora-enriched region. Alternatively, kMT detachment may result from the intrinsic weak resistance of the kinetochore attachment to the assembling kMT ends against a minus end-directed load. In support of the latter possibility, a XMAP215/Dis1 family kinetochore component in budding yeast, Stu2, attached to assembling MT ends can withstand a tensile force of ~4 pN under a plus end-directed load [106], while *Xenopus* XMAP215 can withstand a force of only ~1 pN under a minus end-directed load [107].

Oscillation-dependent attachment elimination could also be responsible for the elimination of merotelic attachments of a single chromatid (Figure 4b, Mitosis, merotelic). Coordinated, cohesin-dependent oscillation of sister centromeres could eliminate merotelic attachments in mitosis in the same way that a coordinated oscillation of homologous chromosomes eliminates bi-oriented attachments of sister centromeres. Supporting this idea, a loss of sister chromatid cohesion has been shown to cause merotelic attachment [108]. In this scenario, the elastic kinetochore/centromere structure, which is important for various kinetochore functions including kinetochore–MT interaction, SAC regulation, and force generation [44,109,110,111,112,113], may be crucial for the elimination of merotelic attachments. Upon merotelic attachment, stretching of the kinetochore and centromere chromatin repositions the bi-oriented attachment sites so that they face in opposite directions on opposing sides [32,114]; this positioning may contribute to the efficient application of a minus end-directed load to improper attachment sites during centromere oscillation.

Notably, this model can also account for the elimination of erroneous mono-oriented attachments (Figure 4b, mono-oriented). During poleward movement of chromosomes, the switching of kMT assembly/disassembly dynamics could cause the application of a minus end-directed load to assembling kMT ends, resulting in kMT detachment. When this poleward chromosome movement is blocked, by physical intervention with an ultra-fine needle, for example [4], kMT detachment may not occur due to a lack of the minus end-directed load.

In the oscillation-dependent model, a decrease in overall tension is not the direct cause of attachment elimination, but a tension decrease is still crucial for attachment elimination. There is no doubt that tension increases as the number of bi-oriented attachments increases [115,116]. A gradual elevation of tension may incrementally decrease the phosphorylation levels of kinetochore components (e.g., see References [25,116,117,118]). The incremental dephosphorylation of kinetochores may gradually increase the MT binding affinity of kinetochores, as seen for dephosphorylation-dependent increase in MT binding affinity of Ndc80 [97]. An alternative but not mutually exclusive possibility is that increased tension alters the phosphorylation state of kinetochores. Kinetochore phosphorylation is regulated by PP1 phosphatases and Aurora B kinase [26,27,28,29]. Antagonistic actions of these enzymes perhaps maintain the phosphorylation state of kinetochores during metaphase, but once tension exceeds some threshold, kinetochores become irreversibly and completely dephosphorylated [119,120,121,122]. In either case, upon establishment of bi-oriented chromosome attachment, tension converts the dynamic metaphase-like attachment into an anaphase-like, non-eliminable attachment that allows only poleward movements, linking attachment establishment with the metaphase-to-anaphase transition. In the model, a decrease in tension would increase kinetochore phosphorylation, making the kinetochore more susceptible to attachment correction. This would account for the promotion of attachment correction by reduced tension [123,124].

## 6. Future Direction

The mechanisms of attachment establishment and metaphase centromere oscillation have been extensively studied at the molecular level. However, the relationship between attachment establishment and centromere oscillation is poorly understood. It is apparent that attachment establishment must be studied in the context of centromere oscillation because these processes take place concurrently. The recent study of *S. pombe* suggested that centromere oscillation contributes to attachment elimination. However, whether centromere oscillation directly contributes to attachment elimination remains to be elucidated. In *dam1∆* cells of *S. pombe*, for instance, the lack of centromere oscillation may result in loss of tension at the kinetochores, thereby impairing selective elimination of erroneous attachments, and centromere oscillation may not directly contribute to elimination. To validate this model further, it will be important to understand the details of the relationship between attachment elimination and centromere oscillation. In this regard, the effects of phospho-mutations at the Aurora B–phosphorylation sites of kinetochore components on attachment correction and centromere oscillation should be intensively studied, as Aurora B regulates these processes. Furthermore, it would be inevitable to investigate whether kinetochore attachment to the assembling MT ends is indeed weak against the minus end-directed load, using single-molecular techniques. Undoubtedly, understanding the relationship between attachment establishment and centromere oscillation will contribute to our understanding of faithful chromosome segregation, which is essential for the stability of eukaryotic genomes.

## Figures and Tables

**Figure 1 ijms-22-03174-f001:**
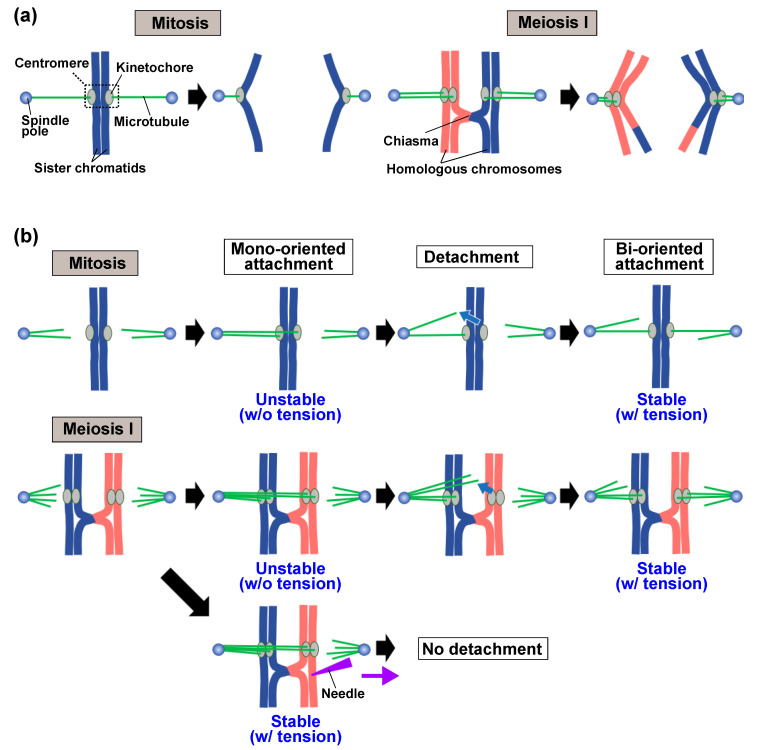
Chromosome segregation in mitosis and meiosis I, and correction of chromosome attachment to the spindle. (**a**) Chromosome attachment to the spindle and chromosome segregation during mitosis and meiosis I. In mitosis, sister chromatids are attached to both spindle poles by microtubules (MTs) that extend from the poles via kinetochores assembled on centromeres, resulting in segregation of the sister chromatids to opposite poles (mitosis). In meiosis I, homologous chromosomes linked by chiasmata are attached to opposite poles, resulting in segregation of the homologous chromosomes (meiosis I). (**b**) The relationship between elimination of improper attachments and tension generated at the kinetochores. When both sister chromatids in mitosis or homologous chromosomes in meiosis I are attached to the same pole, insufficient tension is generated and attachment is unstable (mono-oriented attachment). As a result, chromosomes become detached from the spindle (detachment). The chromosomes again interact with the spindle MTs. When they are bi-oriented, tension is generated, resulting in stabilization and retention of the attachments (bi-oriented attachment). When anti-poleward forces are exerted on mono-oriented homologous chromosomes by an ultra-fine needle, chromosome detachment does not take place (meiosis I, lower row). Blue arrows indicate MT detachment from kinetochores, the purple arrow indicates the force exerted by the needle, and black arrows indicate transition of the attachment state. For simplicity, only one kMT attached to each single kinetochore is shown.

**Figure 2 ijms-22-03174-f002:**
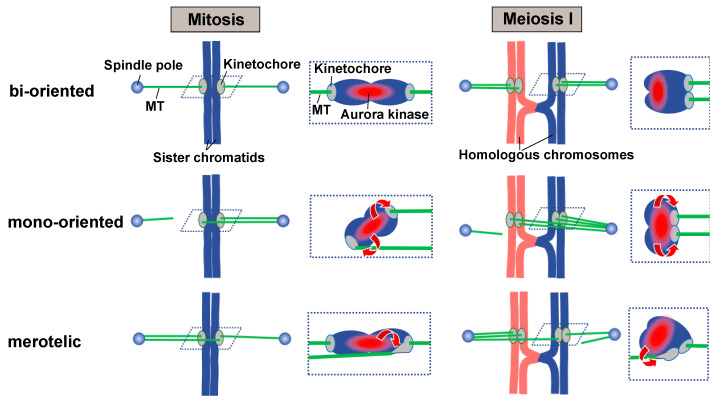
The current model of Aurora B–dependent elimination of improper attachments. Aurora B kinase is enriched at the inner centromere region, where sister chromatids associate with one another (red regions in dotted boxes). Bipolar attachment of sister chromatids in mitosis, or homologous chromosomes in meiosis I, causes stretching of centromeric chromatin and spatial separation of the kinetochore from the Aurora B–enriched region (bi-oriented). When sister chromatids or homologous chromosomes are attached to the same pole in mitosis or meiosis I, respectively, centromeric chromatin does not stretch, and the kinetochores are juxtaposed with the Aurora B–enriched region (mono-oriented). This juxtaposition causes phosphorylation of kinetochore components and elimination of attachments. When a single kinetochore is concomitantly attached to opposite poles in mitosis, the improper attachment site of the kinetochore approaches the Aurora B–enriched region (mitosis, merotelic). Similar spatial proximity of improper attachment sites to the Aurora B–enriched region occurs when associated sister kinetochores are attached to opposite poles at meiosis I (meiosis I, merotelic). Dotted boxes show kinetochore–MT interactions, and the predicted positions of kinetochores and the Aurora B–enriched region. Red arrows indicate Aurora B–dependent kinetochore phosphorylation. For simplicity, only one kMT attached to each single kinetochore is shown.

**Figure 3 ijms-22-03174-f003:**
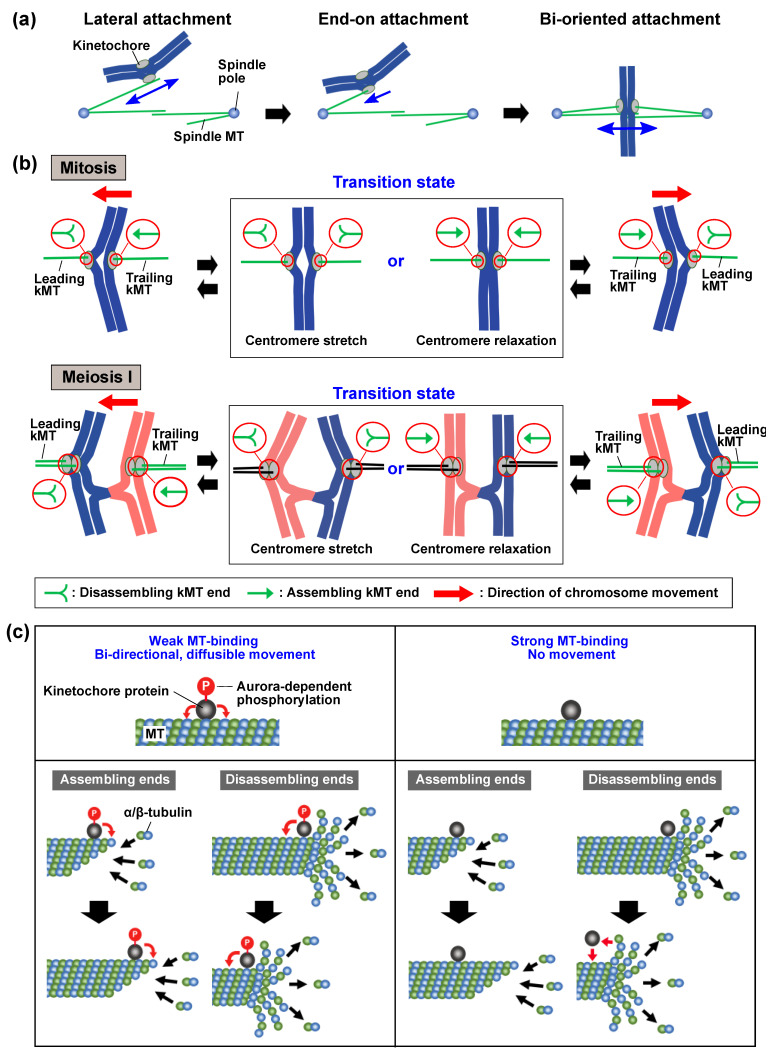
The process of attachment establishment and the mechanism of oscillation of bi-oriented chromosomes. (**a**) The process of chromosome attachment to the spindle in mitosis. Sister chromatids initially interact laterally with MTs extending from the spindle pole (spindle MT) and repeat poleward and anti-poleward movements (lateral attachment). MT shortening promotes the interaction of the kinetochore with the end of the MT (end-on attachment). After end-on attachment of one kinetochore, the other kinetochore interacts with MTs extending from the other pole, resulting in bi-oriented attachment of sister chromatids (bi-oriented attachment). Blue arrows show movement of chromosomes. (**b**) MT dynamics and oscillation of bi-oriented chromosomes in mitosis and meiosis I. During poleward movements of bi-oriented chromosomes, kinetochore-interacting MTs (kMTs) that extend forward of moving chromosomes (leading kMTs) undergo disassembly and shortening, generating pulling forces that drive chromosome movements. The kMTs extending rearward of the chromosomes (trailing kMTs) undergo assembly and elongate. Initial switching of assembly/disassembly dynamics of kMT is induced stochastically or by length-dependent MT disassembly factors. Subsequent changes in tension across the bi-oriented centromeres induce switching of kMT dynamics and the reversal of chromosome movements. Upon reversal, two transition states arise: In one state, both kMTs undergo disassembly and shorten, increasing tension and causing centromeric chromatin, to stretch (transition state, centromere stretch); in the other state, both kMTs undergo assembly and elongate, decreasing tension and allowing centromeric chromatin to relax (transition state, centromere relaxation). Circles indicate the assembly/disassembly states of kMT ends. For simplicity, only one kMT attached to each single kinetochore is shown. (**c**) MT-binding properties (upper panels) and MT end-tracking activity (lower panels) of kinetochore proteins that are phosphorylated (left panels) or non-phosphorylated (right panels) by Aurora B kinase. Red arrows indicate movements of the kinetochore proteins. The phosphorylated kinetochore proteins can move toward the minus or plus ends, and follow both assembling and disassembling ends. The unphosphorylated kinetochore proteins do not move on MTs and cannot follow assembling MT ends, but MT disassembly dependent release of kinetochore proteins from MT ends enables the kinetochore proteins to follow disassembling MT ends through MT dissociation and association processes.

**Figure 4 ijms-22-03174-f004:**
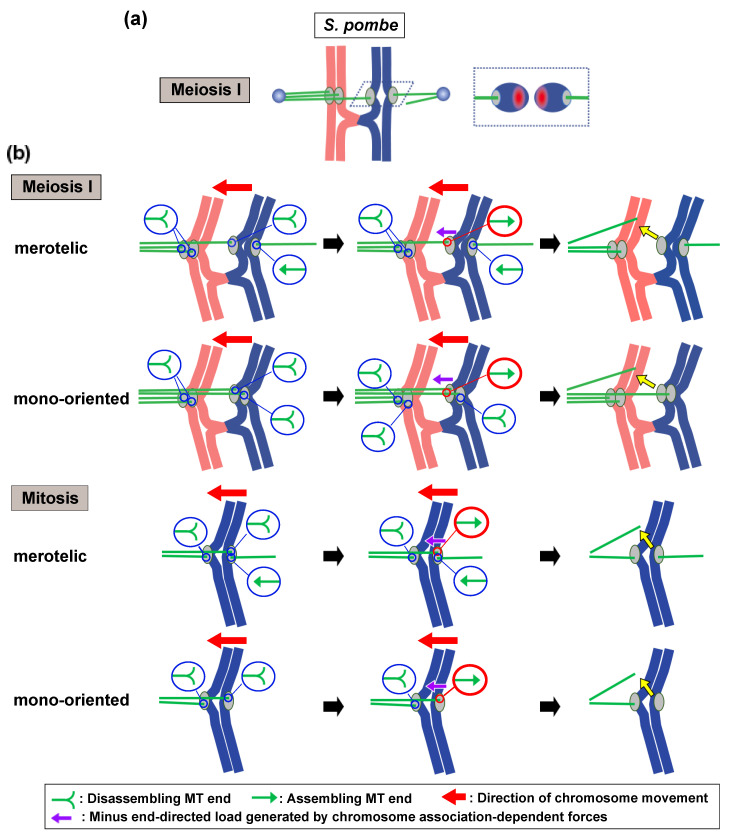
A problem with the tension-dependent spatial separation model and attachment elimination by chromosome oscillation. (**a**) A problem with the tension-dependent spatial separation model at meiosis I in fission yeast. Because sister centromeres separate from each other after their bi-oriented attachment, orientation of kinetochores/centromeres easily changes. Consequently, erroneous attachment sites do not approach the Aurora B–enriched region, preventing the phosphorylation of kinetochore components by Aurora B. (**b**) The model for oscillation-dependent elimination of improper attachments at meiosis I (meiosis I) and in mitosis (mitosis). During poleward movements of erroneously attached chromosomes, kMTs extending forward shorten, whereas those extending rearward elongate. Unsynchronized switching of assembly/disassembly dynamics of kMTs generates occasions, in which chromosome association-dependent pulling (purple arrows) exerts a minus end-directed load on assembling kMT ends (red circles), leading to kMT detachment. Such situation cannot occur at proper attachment sites (see Figure 3b). Meiosis I/merotelic: simultaneous bi-oriented attachments of one sister chromatid pair and homologous chromosomes; meiosis I/mono-oriented: mono-oriented attachment of homologous chromosomes; mitosis/merotelic: simultaneous bi-oriented attachments of a single chromatid and both sister chromatids; mitosis/mono-oriented: mono-oriented attachment of both sister chromatids. Circles indicate the assembly/disassembly states of kMT ends. For simplicity, only one kMT attached to each single kinetochore is shown.

## Data Availability

Data sharing not applicable.

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
