# Peer review of "Shake It Off: The Elimination of Erroneous Kinetochore-Microtubule Attachments and Chromosome Oscillation"

_ijms, 2021, doi:10.3390/ijms22063174_

Round 1

Reviewer 1 Report

The peer-reviewed manuscript  «Shake it off: the elimination of erroneous kinetochore-microtu bule attachments and chromosome oscillation»  is a revised version of a paper I reviewed earlier. The new version differs significantly from the previous version for the better. The author expanded the scope of the discussed literature, added a discussion of his last paper, which added novelty.

The review is devoted to a very interesting and important topic. At the beginning of metaphase of mitosis and metaphase 1 of meiosis, incorrect attachment of microtubules to kinetochores is observed with a high frequency, and the vast majority of these incorrect attachments are corrected by the beginning of anaphase. First, the author discusses one of the most frequently discussed mechanisms for this correction - the mechanism associated with different tension  between sister kinetochores and with local concentration of Aurora B kinase. The author shows that this explanation is not sufficient, primarily to explain the removal of incorrect attachments of microtubules in meiosis. The author views chromosome oscillation as an important mechanism contributing to the removal of incorrect attachments.

The article has clear logic and is well illustrated with diagrams. At the same time, it seemed to me that the presentation of some aspects is superficial. So, in order to understand the forces that act on the chromosome and lead to oscillations, I had to attract additional literature. The advantage of the stylistics chosen by the author for the schemes is the ability to illustrate all the issues raised in the article on similar schematically depicted chromosomes. The downside is that there are no cohesins on the diagrams, which can be misleading about the regions of chromosomes that are in contact, in which regions there is the tension. The shape of the depicted meiotic bivalents also does not correspond to the shape of the chromosomes to which the forces described by the author are applied.

Nonetheless I believe that the manuscript presented now can be published in the International Journal of Molecular Sciences.

Reviewer 2 Report

Author did significant modifications from original manuscript.  This revised manuscript was well written and no more concerns.

Reviewer 3 Report

The manuscript provides essential information regarding the area of cell biology related to kinetochore-MT interaction. The narrative benefited after corrections according to the reviewers' suggestions. The only thing I'd like to advise: if Fig 1. (a) in the "mitosis" panel right chromatid with centrosome is situated too close to the "meiosis" section of the picture, wrongly becoming the part of it. 

This manuscript is a resubmission of an earlier submission. The following is a list of the peer review reports and author responses from that submission.

Round 1

Reviewer 1 Report

The article highlights a modern understanding of mechanisms responsible for proper kinetochore attachments during metaphase alignment and anaphase transitioning. The main focus made for microtubule dynamics and the role of Aurora B kinase in the establishment of bioriented attachment chromatids to the spindle. That’s an informative manuscript, well-structured and illustrated, and I enjoined the reading of it.

However, the author could consider inserting several minor language corrections:

Using the determiners “the” might improve overall readability of some sentences:

10                                                        the survival

15, 50, 87,  252, 285, 305                   the elimination  

31                                                        the division

33, 37                                                  the formation

95, 97                                                  the establishment

111                                                      the subsequent gathering

150, 154                                              the interaction

160                                                      the reversal  

177                                                      the tension

231                                                      the selection

37 remove “a”                                  and subsequent equational segregation

82 remove “to”                                   may not be so simple

233           consider to remove tautology “and  moreover” or put an additional comma “and, moreover,”

255, 268                                              use lower case (a) and (b)

Reviewer 2 Report

This review picked one of important topics in cell division machinery, which discussed chromosome oscillation underlying establishment of biorientation. Chromosome movements during cell division are largely depends on force generated by microtubules, especially force at kinetochores, which is macro-molecular protein complex on chromosomes. Oscillation of chromosomes as well as sister kinetochore pairs thought to be important for ensuring fidelity of chromosome segregation. It is very attractive to discuss this important topic; however, unfortunately I seriously disappointed after reading this review and I could not recommend it for publication in International Journal of Molecular Sciences without significant revision.

Major concerns

  1. Author lacked significant amount of important original papers to cite. In this kind of review, it is inappropriate to cite a review paper, the author must respect original works.
  2. Related above 1, there are many important uncited papers, even not in the review author cited.
  3. Author mentioned multiple times, “many studies, previous work…etc” without citation.
  4. In the context of force on chromatin in cell division, there are two different types of force; force between sister kinetochores (contributes for inter-kinetochore stretch) and force within a single kinetochore (contributes probably both inter- and intra-kinetochore stretch). Both forces are not completely independent but some case, those work independently. In this review, unfortunately, author confused both forces or at least current version makes reader confused for above two types force. There were many important papers showed or discussed contribution of force within a single kinetochore for fidelity of chromosome segregation, however, author did not discuss at all. Some discussion in Figure 4, but no any important papers were cited and discussed their findings.
  5. What is the mean of centromere oscillation? In budding yeast, they have point centromeres. Centromere oscillation was used as sister-kinetochore oscillation. However, it is known that centromere chromatin distributed entire inter-centromere (between sister kinetochore pairs) in human. CENP-B, which binds centromere DNA, localized entire inter-centromere in both light and immuno-electron microscopic observation. Author needs to define what is the centromere oscillation as well as clearly separate from force within a single kinetochore.
  6. Lacking discussion of Aurora A at the pole in chromosome oscillation (movements) for error correction.
  7. Why did author used “pre-anaphase” instead of before metaphase? The definition of metaphase is the time establish biorientation and just before anaphase. Is there any specific reason to use pre-anaphase? Attachment error corrections should occur during prometaphase before metaphase since it is known that microtubule-kinetochore attachments make stronger at metaphase as well as dephosphorylation of Ndc80.
  8. Author must need to discuss Aurora B gradient model with PP1 as well as intra-kinetochore stretch in addition to sister kinetochore oscillation.
  9. Lacking discussion about papers, which Aurora B artificially removed from inner centromere.
  10. L347, although author lacked citation, this statement was supported only situation using artificial phospho- and dephospho-mimetic mutants of Ndc80 complex or other kinetochore proteins. Aurora B inhibitor treatment did not support such a strong conclusion. Phospho- or dephospho-mimetic mutants are artificial situation, and usually it showed stronger phenotypes. For example, MT binding site of Ska is not essential for cell division in C. elegans but artificial phsopho-mutant did not support chromosome segregation. In addition, dephsopho-mimetic Ndc80 in human showed much stronger phenotypes than Aurora B inhibitor treatments.
  11. Author stated that monopolar spindle has no tension. It is true in force between sister kinetochores, however, it is completely wrong in force at a single kinetochore. There is significant force at a single kinetochore, otherwise chromosomes could not be moved to a pole. I believe author is well-understood those things, but current version of this review makes significant mis-leading for readers.
  12. Recent budding yeast paper showed that random dynamic instability of 16 MTs can make fluctuation of sister kinetochore clusters. Author should mention that both coordinated and potentially non-coordinated dynamic instability can establish both oscillation and fluctuation of sister kinetochore pairs.

Minor concerns

  1. L202, “These findings…”. I do not think cited papers support this conclusion. Citrated papers showed importance of Aurora B activity in error correction but there are many other important works to show Aurora B functions in context of tension.
  2. Related to above or major concerns, many statements lacked citation as well as the review missed or ignored important citations as well as critical components, which should be discussed in the topic discussed in this review.
  3. L118, Author needs to clearly mention Dam1 ring like structure is observed only in vitro at this point.

Reviewer 3 Report

The paper under consideration is a review on a relevant and interesting topic, the author sets the task to understand the current state of the problem of the relationship between establishment of chromosome attachment to the spindle and chromosome oscillation. Such a statement of the problem seems to me quite worthy of a review article.

The review is written in good language, with considerable logic, has a large number of illustrations, but in the current it looks more like a chapter in a textbook rather than a scientific review paper.

A scientific review paper usually involves a synthesis of current knowledge at a new level. It is noteworthy that only 13 references out of 82 refer to articles written in the last 5 years and only one reference  later than 2017. Thus, all the latest achievements in the field considered by the author are completely beyond the scope of the review. I got the impression that the article presented was written several years ago as a chapter for a book. In particular, the author often says "recent studies show ...", citing articles from almost a decade ago.

All the figures were drawn very soundly, they would be brilliant illustrations for a textbook, but, unfortunately, in the present form they do not carry new information. For each picture, one can find similar pictures in modern reviews, even with much more details (see for example, the papers: Vladimirou  et al., Chromosome Res. 2011, Amaro  et al., Nat Cell Biol. 2010. Thomas et al.,  Biochemical Journal 2017, Funabiki, Current Opinion in Cell Biology, 2019).  In the text, the problem is considered at a more complex and interesting level, but it turns out that more complex issues are described very briefly and are not illustrated. The names of the proteins are introduced, but only for some of them the mechanisms of action are described. Therefore, to understand the issue, one has to use additional sources.

Chromosome oscillation is the main topic of the paper, but many questions related to studies of oscillation mechanisms are beyond the scope of the review. Many works have been devoted to modeling this process and describe in more details the factors responsible for it (see for example, Klemm  et al., Mol Biol Cell. 2018, Su et al.,  Cell Rep., 2016,  Cassimeris et al.,  Cell Motil Cytoskeleton. 2009). It would be appropriate to span the discussion of this in the article.

The review has a good and clear structure corresponding to the stated topic. It seems to me that the author needs to revise the review, adding more up-to-date data on the state of the issue, and to specify in more details the mechanisms that are the main topic of the review. Then the author’s concept will become more interesting and understandable.